# Biodegradation Potential of Polyethylene Terephthalate by the Two Insect Gut Symbionts *Xanthomonas* sp. HY-74 and *Bacillus* sp. HY-75

**DOI:** 10.3390/polym15173546

**Published:** 2023-08-25

**Authors:** Jong-Hoon Kim, So-Hye Lee, Byeong-Min Lee, Kwang-Hee Son, Ho-Yong Park

**Affiliations:** Microbiome Convergence Research Center, Korea Research Institute of Bioscience and Biotechnology, Daejeon 34141, Republic of Korea; kjh1018@kribb.re.kr (J.-H.K.); sohye2202@kribb.re.kr (S.-H.L.); ghsaor@kribb.re.kr (B.-M.L.); sonkh@kribb.re.kr (K.-H.S.)

**Keywords:** polyethylene terephthalate, biodegradation, insect gut symbiont, plastic waste management

## Abstract

Polyethylene terephthalate (PET) is a plastic material that is widely used in beverage bottles, food packaging, and other consumer products, which is highly resistant to biodegradation. In this study, we investigated the effects of two insect gut symbionts, *Xanthomonas* sp. HY-74 and *Bacillus* sp. HY-75, during PET biodegradation. Both strains degraded PET-containing agar plates, and the sole nutrition source assay showed that HY-74 had different degradation rates depending on the presence of specific carbon and nitrogen sources, whereas HY-75 exhibited comparable degradation across all tested conditions. The two strains biodegraded the PET film with 1.57 ± 0.21% and 1.42 ± 0.46% weight loss after 6 weeks, respectively. Changes in the morphology and structure of the PET films, such as erosion, scratching, and surface roughening, were determined using scanning electron microscopy (SEM). Further, the two strains biodegraded PET powder, broke it into its degradation products, and changed the surface functional groups. This is the first study to investigate the biodegradation of PET by Hymenoptera gut-derived microbes and offers promising insights into the potential applications of insect gut symbionts in PET waste management.

## 1. Introduction

Plastics are widely used in human life because of their utility and convenience; however, they cause serious environmental problems. Plastic waste has recently attracted worldwide attention, and many countries have enacted laws and countermeasures. The recycling rate is approximately only 9% despite the fact that global plastic consumption has reached 460 million tons, quadrupling over 30 years [1]. Only 19% of plastics are incinerated, generating toxic products, whereas the remaining 72% are discarded in the environment [2]. Therefore, biodegradation strategies are becoming more crucial than incineration and chemical strategies.

Polyethylene terephthalate (PET) is a widely used plastic material that is commonly used in beverage bottles, food packaging, and other consumer products. It is a thermoplastic polymer that is composed of repeating units of terephthalic acid (TPA) and ethylene glycol (EG), which are linked together through ester bonds. It is lightweight and convenient, making it a popular choice for sustainable packaging. However, PET’s physical resistance to degradation contributes to it being a major source of plastic pollution as it can take hundreds of years to decompose in the environment [3]. Therefore, researchers are exploring ways to biodegrade PET using microorganisms, enzymes, and other methods to reduce its environmental impact.

Scientists have discovered several bacteria that are effective plastic-degrading agents. For example, in the case of PET degradation, *Ideonella sakaiensis* was discovered in 2016 and is capable of breaking down PET into its monomers [4]. Other bacteria, such as *Bacillus* sp. [5,6,7,8], *Pseudomonas* sp. [9,10], *Thermobifida fusca* [11], and *Rhodococcus ruber* [12], have been found to degrade PET through the production of various enzymes. Plastic-degrading microorganisms are found in diverse environments, including soil, seawater, and plastic waste. In the case of insects, their potential role in breaking down plastic was first discovered when they consumed packaging materials [13]. Since then, many studies have been conducted to identify insect gut-related microbes with rare plastic-degradation properties. Yang et al. [14] demonstrated that gut-associated bacteria in mealworms (*Tenebrio molitor*) could effectively degrade various types of plastics. Similarly, other studies have shown that the gut microbes of waxworms (*Galleria mellonella*) can degrade various types of plastics, including low-density polyethylene and polypropylene (PP) [15,16]. However, most studies have been limited to specific insects and plastic materials, and PET biodegradation by insect-derived microbes has not yet been reported. Therefore, there is an urgent need to secure microbial resources based on biodiversity and promote scientific advancement in PET biodegradation.

In this study, two PET-degrading bacterial strains, *Xanthomonas* sp. HY-74 and *Bacillus* sp. HY-75, were isolated from the intestines of two Hymenoptera species: *Xylocopa appendiculata* and *Eumenes decoratus*, respectively. The PET degradation abilities of HY-74 and HY-75 were evaluated using various physicochemical techniques, including weight loss measurements, scanning electron microscopy (SEM), Fourier transform infrared spectroscopy (FTIR), and high-performance liquid chromatography (HPLC).

## 2. Materials and Methods

### 2.1. Plastic Materials

The amorphous PET film was purchased from Goodfellow (Huntingdon, UK), cut into pieces (20 mm × 10 mm × 0.25 mm), and weighed. After soaking the film in 70% (*v*/*v*) ethanol overnight, the PET film was air-dried on a clean bench. Commercial PET powder (particle size <300 μm) was purchased from Goodfellow (Huntingdon, UK), soaked in 70% (*v*/*v*) ethanol, and air-dried on a clean bench.

### 2.2. Screening, Isolation, and Identification of PET Degrading Bacteria

The adult bees, *Xylocopa appendiculata* and *Eumenes decorates*, collected from the mountain Myongdo-bong (Jinan-gun, Jeollabuk-do, Republic of Korea), were first washed with 70% (*v*/*v*) ethanol to remove contaminants on the body surface before being washed twice with sterilized water. The digestive tract was dissected, and the intestinal contents were carefully recovered. The intestinal contents were diluted in phosphate-buffered saline and spread onto a double-layer agar plate containing 0.1% (*w*/*v*) PET, as described by Charnock et al. (2021) with minor modifications. The top layers were 1.5% agar and 0.1% PET powder dissolved in 40 mL dimethyl sulfoxide (Duchefa, Haarlem, Netherlands), and Reasoner’s 2A agar (R2A; MBcell, Seoul, Republic of Korea) was used as the bottom layer in the PET agar plate. After incubation at 30 °C for 14 days, the bacteria that formed a translucent halo around the areas of bacterial growth were selected for further study.

The two strains were assessed for their extracellular enzyme activity as described by Bhagobaty and Joshi [17] with minor modifications. The activated bacterial cultures were streaked onto each R2A agar plate containing a specific substrate and incubated at 30 °C for 3 days. The ability to produce protease and polycaprolactone (PCL) degrading enzyme was qualitatively confirmed by observing transparent halo zones surrounding the colonies on the R2A agar containing 2% (*w*/*v*) skim milk and 0.1% (*w*/*v*) PCL, respectively. The extracellular lipase activity was qualitatively determined using R2A agar with 1% (*w*/*v*) Tween 80 and 0.01% (*w*/*v*) CaCl_2_ and confirmed by the formation of crimson dots around the colonies, which were further measured.

Strain identification was performed using poly chain reaction amplification and nucleotide sequence analysis of the 16S rRNA gene. Genomic DNA was extracted using a DNA extraction kit (Bioneer, Daejeon, Republic of Korea) according to the manufacturer’s guidelines. A partial 16S rRNA sequence from the isolated strain was amplified for phylogenetic identification using the universal primers 24F (5′-AGAGTTTGATCCTGGCTCAG-3′) and 1492R (5′-AAGTCGTAACAAGGTAACC-3′). The 16S rRNA sequences of closely related strains were retrieved from the National Center for Biotechnology Information (http://www.ncbi.nlm.nih.gov/GenBank/index.html; access on 11 September 2022) and aligned using CLUSTAL X [18]. The molecular phylogeny of 16S rRNA was inferred using the neighbor-joining method in MEGA software version X.

### 2.3. Sole Nutrition Source Assay

The ability of HY-74 and HY-75 to utilize PET as their sole nutrient source was assessed. These strains’ growth and PET-degradation capabilities were investigated using a minimal salt medium (MSM; 0.1% NH_4_NO_3_, 0.07% K_2_HPO_4_, 0.02% KH_2_PO_4_, 0.02% CaCl_2_·2H_2_O, 0.005% KCl, 0.001% FeSO_4_·7H_2_O, 0.001% ZnSO_4_·7H_2_O, 0.001% MnSO_4_·7H_2_O, 15 g agar, and 1 L water) with PET as the sole nutrient source. The PET plates containing double-layered agar were prepared as described above. Four variations of MSM were prepared as the bottom layers: MSM, MSMG (0.5% glucose supplementation), MSMN (0.5% ammonium sulfate supplementation), and MSMGN (0.5% glucose and 0.5% ammonium sulfate supplementation). A total of 10 μL of activated cultures of HY-74 and HY-75 was added to each prepared plate and incubated at 30 °C for 4 weeks. After 4 weeks, translucent halos around the areas of bacterial growth in each medium were monitored and calculated using image J version 1.53t (US National Institute of Health, MD, USA). HY-74 and HY-75 were collected from each agar plate by scraping the surface with a cell scraper (SPL Life Science Inc., Pochen, Republic of Korea), and the total cell dry weight was measured to evaluate growth in each medium.

### 2.4. Degradation of PET Film

A total of 100 µL of HY-74 and HY-75 bacterial suspensions (the mid-log phase) was inoculated into each 10 mL of MSM, R2A, and LB (BD Difco, Franklin Lakes, NJ, USA) media in 50 mL Erlenmeyer flasks containing pre-weighed pieces of PET film (as described in Section 2.1), and the flasks were incubated on a rotary shaker (180 rpm) at 30 °C for 6 weeks, 8 mL of incubated media was replaced with 8 mL of fresh media every week. A negative control without bacterial inoculation was maintained under the same conditions. After incubation, each film was rinsed with 1% SDS and then thoroughly washed with distilled water five times to remove residual cells. The completely air-dried film was weighed and collected to observe the changes in the surface morphology and polymer bond formation. The experiments were performed in triplicate under the same conditions. For the SEM observations, the washed PET films were coated with gold using a sputter coater (Q15ORS, Quorum, East Sussex, UK) and analyzed using the FEI Quanta 250 FEG (FEI, Hillsboro, OR, USA). A PET film without inoculation was used as a negative control. The FTIR spectra of the PET films treated with or without bacteria were recorded at wavenumbers ranging from 400 to 4000 cm^−1^ (4 cm^−1^ resolution) using a Nicolet iS50 infrared spectrometer (Thermo Fisher Scientific Instrument, Waltham, MA, USA).

### 2.5. PET Powder Degradation Assay

HY-74 and HY-75 were cultured on a rotary shaker at 30 °C for 4 weeks in 250 mL Erlenmeyer flasks containing 50 mL of R2A with 1% (*w*/*v*) PET powder (<300 μm). After incubation, the supernatants were centrifuged at 13,000× *g* for 15 min at 4 °C and subsequently filtered using a 0.22 μm filter (Millipore, Burlington, MA, USA). The supernatant was freeze-dried using a freeze dryer (Alpha 1–4 LD plus, Martin Christ, Osterode, Germany), extracted with methanol, and filtered using a 0.22 μm pore filter.

An Agilent 1260 Infinity II Quaternary LC (Agilent, Santa Clara, CA, USA) with a Brownlee SPP C18 column (4.6 mm × 50 mm × 2.7 μm, PerkinElmer, Waltham, MA, USA) was used for HPLC analysis. Methanol and 50 mM phosphoric acid were used as the mobile phases with a gradient of 20% methanol for 0–2 min and 40% methanol for 12 min. The column was maintained at 40 °C, and the flow rate was 1 mL/min. The wavelength of the UV detector was set at 254 nm. Bis(2-hydroxyethyl) terephthalic acid (BHET) (Sigma Aldrich, St. Louis, MO, USA), mono(2-hydroxyethyl) terephthalic acid (MHET) (Advanced ChemBlocks Inc., Hayward, CA, USA), and TPA (Sigma Aldrich) were dissolved in methanol.

### 2.6. Statistical Analysis

One-way analysis of variance was performed using SPSS software (version 24; SPSS, Inc., Chicago, IL, USA). To compare the mean values, Scheffé’s method was employed, and *p*-values less than or equal to 0.05 were deemed statistically significant.

## 3. Results

### 3.1. Isolation and Identification of PET Degrading Strain

Two bacterial strains isolated from the intestines of two Hymenoptera, *Xylocopa appendiculata* and *Eumenes decorates*, had a high degree of transparent halo zones on the test plate containing 0.1% (*w*/*v*) PET (Appendix A, see in Appendix A). According to phylogenetic profiling based on the partial 16S rRNA region, the HY-74 strain was most closely related to the *Xanthomonas sontii* strain PPL1 (NQYO01000058) with 99.92% 16S rRNA nucleotide sequence similarity (Figure 1A). The HY-75 strain was most closely related to the *Bacillus siamensis* strain KCTC13613 (AJVF01000043) with 99.57% 16S rRNA nucleotide sequence similarity (Figure 2A). The 16S rRNA nucleotide sequences of the two strains were deposited in GenBank under accession numbers OQ921840 and OQ921871. The two strains were selected as potential candidates for PET degradation and were stored for further studies.

Two bacterial strains were evaluated for the production of extracellular enzymes (Figure 1B and Figure 2B). The results indicated that both strains have the ability to produce extracellular protease, PCL-degrading enzyme (the formation of a transparent halo zone), and lipase (the formation of a calcium complex).

### 3.2. Sole Nutrition Source Assay

The ability of HY-74 and HY-75 to use PET as the sole nutritional source was assessed under four different carbon and nitrogen conditions in MSM using a double-layer PET agar plate (Figure 3). Over the course of 28 days, the HY-74 strain exhibited the following dry cell weight at the four different conditions: (i) 0.18 ± 0.05 g/L with MSM, (ii) 0.62 ± 0.10 g/L with MSMG, (iii) 0.60 ± 0.15 g/L with MSMN, and (iv) 1.36 ± 0.10 g/L with MSMGN. The HY-75 strain displayed the following dry cell weight at the four different conditions: (i) 0.13 ± 0.05 g/L with MSM, (ii) 0.57 ± 0.13 g/L with MSMG, (iii) 0.35 ± 0.08 g/L with MSMN, and (iv) 0.98 ± 0.14 g/L with MSMGN. The growth of the two strains on PET as the sole carbon and nitrogen source was obvious, but the growth was lower than that on minimal medium with an additional carbon and nitrogen source. The two strains exhibited the formation of degradation halos, as monitored by visual inspection. Under the same conditions, the PET degradation ability of the HY-74 strain was significantly higher than that of the HY-75 strain. The halo zones of the HY-74 strain were 319 ± 50 mm^2^ with MSM, 357 ± 63 mm^2^ with MSMG, 356 ± 30 mm^2^ with MSMN, and 287 ± 51 mm^2^ with MSMGN. HY-75 exhibited transparent halo zones of (i) 238 ± 32 mm^2^ with MSM, (ii) 169 ± 27 mm^2^ with MSMG, (iii) 473 ± 66 mm^2^ with MSMN, and (iv) 201 ± 63 mm^2^ with MSMGN.

### 3.3. PET Film Degradation

The ability of the HY-74 and HY-75 strains to degrade the PET film was determined using weight loss measurements, SEM observations, and FTIR analysis (Figure 4). The dry weight of the PET film after six weeks of incubation was significantly lower in all the tested media. In the HY-74 strain, the weight losses of the treated films were the following: MSM (0.087 ± 0.017 mg), R2A (0.197 ± 0.063 mg), and LB (0.196 ± 0.018) (Figure 4E). The HY-75 strain-degraded film contained MSM (0.097 ± 0.018 mg), R2A (0.210 ± 0.029 mg), and LB (0.163 ± 0.041 mg) (Figure 4D). With a weight loss of 1.57 ± 0.21% (for HY-74) and 1.42 ± 0.46% (for HY-75), R2A media showed the highest PET film degradation. The dry weight of the PET film of the untreated sample did not change after six weeks of incubation.

The surface morphological changes of the PET film were analyzed using SEM at 20,000× and 40,000× magnifications after six weeks of incubation (Figure 4A–C). The PET films incubated with both strains showed clear morphological changes due to microbial degradation. Scanning electron micrographs of the PET films treated with both strains showed significant damage to the surface morphology, such as erosion, scratches, and roughness, compared to the control.

The FTIR spectra of the PET film degradation by strains HY-74 and HY-75 showed changes in functional groups accompanied by more subtle changes at other wave numbers (Figure 4F,G). The peaks associated with aromatic C-H stretching (723 cm^−1^), C-O stretching (1097 cm^−1^), C-O-C stretching (1240 cm^−1^), and C=O stretching (1714 cm^−1^) decreased compared to the control. For HY-74, the O-H stretching (3200–3600 cm^−1^) increased, whereas no significant difference was observed for HY-75. Specifically, the C-H stretching (723 cm^−1^) corresponds to the out-of-plane bending vibrations in the aromatic and aliphatic C-H bonds, while the C-O stretching (1097 cm^−1^) indicates the ester linkages between the terephthalate and EG units. Furthermore, the C-O-C stretching (1240 cm^−1^) represents the vibrations in the ester group, and the C=O stretching (1714 cm^−1^) signifies the cleavage of the ester bonds in PET. The increased hydroxyl group (OH) peak (3200–3600 cm^−1^) suggested the formation of OH in the degradation products. These changes indicated that the PET film underwent chemical alterations, resulting in the formation of new functional groups and a modified composition.

### 3.4. Degradation of PET Powder

After 4 weeks of incubation with HY-74 and HY-75, the supernatant was analyzed using HPLC to confirm degradation in the presence of degradation products. In the standard HPLC spectra, the PET degradation products, including TPA, MHET, and BHET, showed clear peaks at 3.08, 5.08, and 6.75 min, respectively. Consistently, the supernatants treated with HY-74 and HY-75 showed TPA peaks at a retention time of 3.08 min after 4 weeks as shown in Figure 5, while MHET and BHET were not detected. The absence of these peaks indicates that they were subsequently hydrolyzed to TPA. No degradation products were detected in the control group.

## 4. Discussion

Over the last decade, there has been growing interest in the plastic degradation capabilities of insect larvae for PP, polystyrene (PS), and polyvinyl chloride (PVC) because herbivorous insects may be a valuable resource for microorganisms that can break down synthetic plastics. Insects that can digest plastics using their gut microbiota are gaining interest for use in bioremediation, although their environmental benefits remain unknown [19]. These microbes exploit the enzymatic pathways of decomposing plant materials to degrade synthetic plastic polymers because of their similar chemical structures [20]. However, most plastics biodegraded by insect gut microbes are polyethylene [21,22,23], PS [24], PVC [25,26], and polyurethane [27,28] with little knowledge available on the gut microbiome of insects capable of degrading PET. In this study, the PET degradation abilities of two insect gut-derived bacterial strains, HY-74 and HY-75, were confirmed using PET-containing agar plates, PET powder, and a PET film. To the best of our knowledge, this is the first report on the isolation and characterization of PET-degrading bacteria of insect intestinal origin.

HY-75 demonstrated differential degradation activities under minimal and nutrient-rich conditions. The nitrogen source appeared to be a crucial factor in promoting PET degradation in HY-74, whereas the presence of glucose as a carbon source seemed to have a suppressive effect on degradation. Differences in the degrading abilities of the different media suggest that HY-74 produces enzymes in response to the limited availability of nutrients. In minimal media, microorganisms often need to synthesize specific enzymes to break down and utilize the limited available resources, whereas in nutrient-rich media, they may not need to produce these enzymes because of the abundance of readily available nutrients [29,30]. Enzyme production is usually subjected to catabolite repression in which the presence of a preferred carbon or nitrogen source in rich media inhibits the expression of certain enzymes [31]. These preferred sources may not be present in MSM without glucose, leading to the induction of enzyme production that utilizes alternative substrates. The degradation activity of HY-74 was similar or comparable for all the different nutritional sources. With controlled carbon and nitrogen sources, the degradation activity was not significantly affected by the specific carbon or nitrogen sources present. The degradation-related enzymes of the HY-74 strain may be constitutively expressed and continuously produced by the organism regardless of the presence of a specific substrate. This finding indicates that HY-74 can efficiently degrade PET regardless of the nutritional source.

Quantitative and visual analyses of the PET film degraded by strains HY-74 and HY-75 indicated that they could also degrade PET film types. After 6 weeks of incubation, the highest degradation rates of PET film for HY-74 and HY-75 were 1.57 ± 0.21% and 1.42 ± 0.46%, respectively. In the scanning electron images, the PET film exhibited disruption and changes in the surface morphology. Although similar observations have been previously reported, HY-74 and HY-75 showed similar or higher weight losses than other bacteria on PET films in a shorter period [8,32,33], while their degradation efficiency was not as high as the whole-cell degradation exhibited by *Clostridium thermocellum* [34]. Further investigation of the factors that influence bacterial growth and PET degradation, including the physical properties of PET, temperature, pH, and nutrient availability, is required to develop effective bioremediation strategies.

Many hydrolases from microorganisms, such as cutinase and esterase, have been discovered and modified to primarily target and break down PET ester bonds. Hydrolase can degrade PET into MHET and BHET or ultimately into TPA and EG. As a result, some microorganisms can use the TPA and EG monomers generated in this degradation process, either by incorporating them into the tricarboxylic acid cycle or transforming them into valuable chemicals that enable the biodegradation and bioconversion of PET. In this study, the bacterial strains HY-74 and HY-75 demonstrated the ability to break down PET into TPA, as confirmed with HPLC. We hypothesized that these microorganisms produce a specific type of hydrolase that degrades PET. However, the comprehensive understanding of the specific enzymes involved and the intricate process of TPA metabolism is limited in this study. Further research, such as determining the specific enzymes involved and elucidating the degradation process of TPA metabolism, is required to understand the subsequent PET biodegradation by HY-74 and HY-75 and their bioconversion steps. The findings of this study provide promising prospects for the use of insect gut bacteria for PET biodegradation. Further research should focus on optimizing the conditions for the growth and activity of the identified PET-degrading microorganisms as well as exploring genetic engineering methods to enhance the efficiency of PET biodegradation. For instance, recent studies have focused on engineering and optimizing PET-degrading enzymes to enhance their performance [35], investigating microbial communities for potential PET degradation [36], and developing genetically engineered whole-cell catalysts, such as *E*. *coli*, expressing PET-degrading enzymes to enhance PET-degradation efficiency [37].

## 5. Conclusions

The PET biodegradable bacteria *Xanthomonas* sp. HY-74 and *Bacillus* sp. HY-75 were isolated from the intestines of Hymenoptera, *Xylocopa appendiculata* and *Eumenes decorates*, respectively. Both strains demonstrated the ability to degrade PET as evidenced by the observed changes in chemical functional groups and a distinctive TPA peak in the HPLC results. Additionally, morphological alterations of the PET film, surface modifications, and PET film weight reduction (1.57 ± 0.21% for HY-74, 1.42 ± 0.46% % for HY-75) were observed after a 6-week period. Further research employing multi-omics approaches and synthetic microbial communities is needed to elucidate the specific enzymes involved and their metabolic pathways, and to optimize the degradation conditions for practical applications in PET waste management. This study highlights the potential of insect-derived bacterial strains for PET biodegradation and highlights the importance of further exploration of their practical applications in mitigating the global plastic pollution crisis.

## Figures and Tables

**Figure 1 polymers-15-03546-f001:**
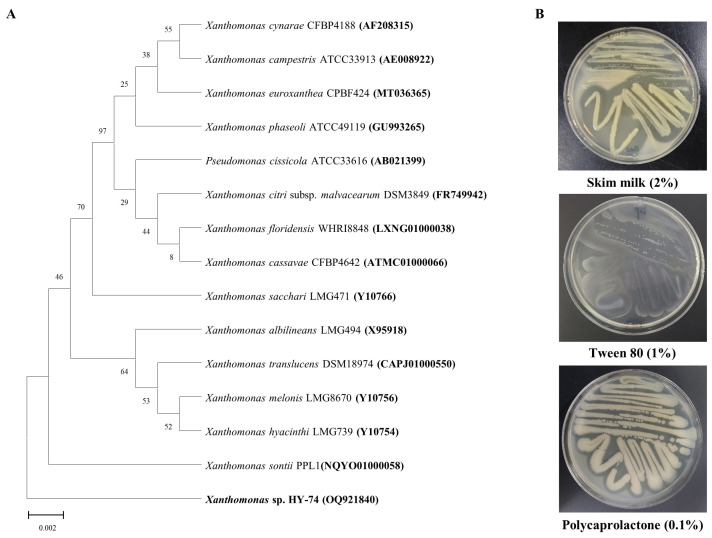
Phylogenetic relationship of the HY-74 strain from the intestine of *Xylocopa appendiculata* based on partial 16S rRNA gene sequencing (**A**). Neighbor-joining phylogenetic tree based on 16S rRNA gene sequences and closely related species constructed using MEGA X software. Numbers at branch nodes indicate the bootstrap percentages of 1000 replications. Extracellular enzymatic activities of the HY-74 strain (**B**).

**Figure 2 polymers-15-03546-f002:**
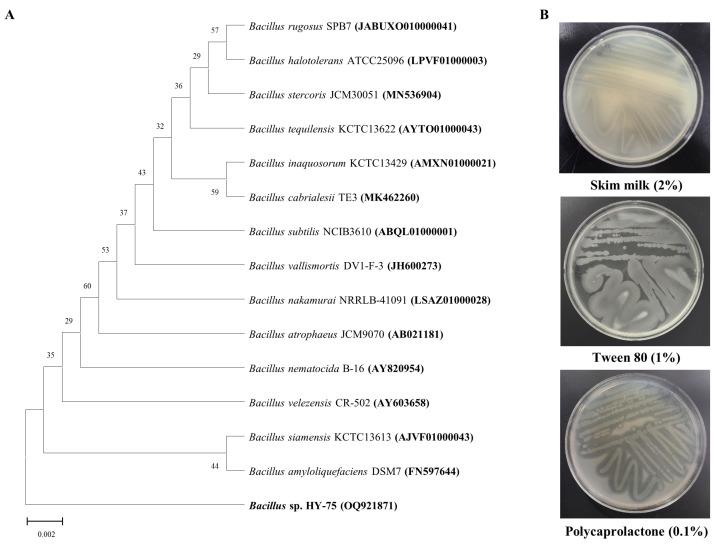
Phylogenetic relationship of the HY-75 strain from the intestine of *Eumenes decorates* based on partial 16S rRNA gene sequencing (**A**). Neighbor-joining phylogenetic tree based on 16S rRNA gene sequences and closely related species constructed using MEGA X software. Numbers at branch nodes indicate the bootstrap percentages of 1000 replications. Extracellular enzymatic activities of the HY-75 strain (**B**).

**Figure 3 polymers-15-03546-f003:**
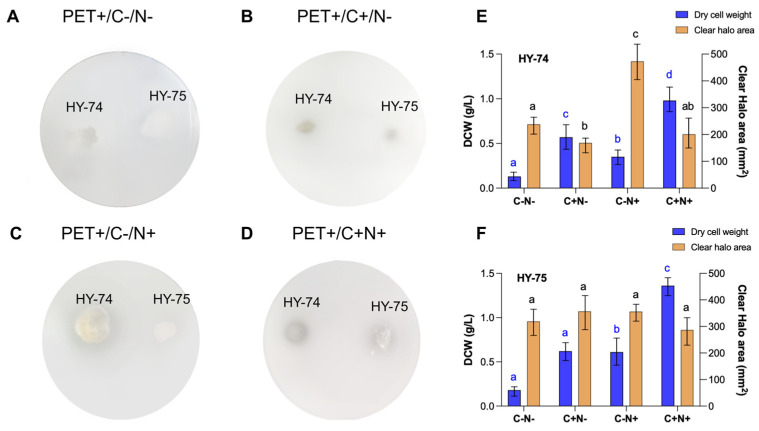
Growth and PET degradation of the HY-74 and HY-75 strains on a PET-containing agar plate as a sole nutrition source. A 0.1% PET-containing minimal salt agar medium was used to grow the strains for 4 weeks. (**A**) Minimal salt medium (MSM); (**B**) MSM supplemented with 0.5% glucose (MSMG); (**C**) MSM supplemented with 0.5% ammonium sulfate (MSMN); (**D**) MSM supplemented with 0.5% glucose and 0.5% ammonium sulfate (MSMGN). Dry cell weight was measured to assess bacterial growth, and PET biodegradation activity was demonstrated by visible clear halo zones surrounding the colonies for HY-74 (**E**) and HY-75 (**F**). Data are presented as means ± SD (*n* = 3). Different letters above the error bars indicate a significant difference by Scheffé’s test (*p* < 0.05).

**Figure 4 polymers-15-03546-f004:**
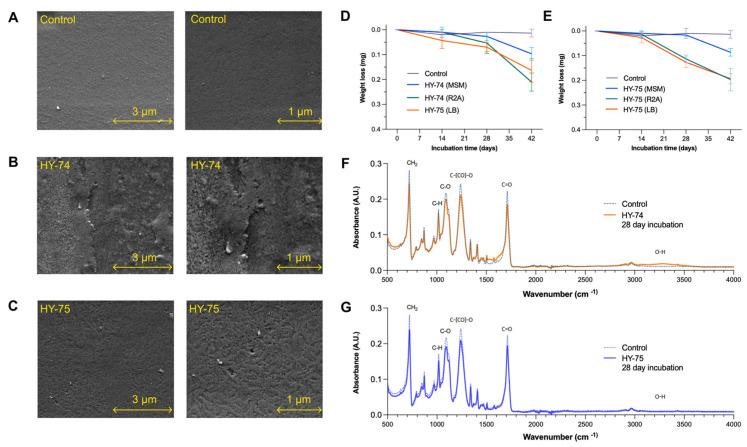
Degradation of PET film by HY-74 and HY-75 strains after 6 weeks of incubation. Scanning electron micrographs of non-treated control films (**A**), films treated with the HY-74 strain (**B**), and films treated with the HY-75 strain (**C**). Time course of degradation of PET film by the HY-74 strain (**D**) and HY-75 strain (**E**) in MSM, R2A, and LB media. Comparison of FTIR spectra of PET film degradation of the HY-74 strain (**F**), HY-75 strain (**G**), and non-treated control. Data are presented as means ± SD (*n* = 3).

**Figure 5 polymers-15-03546-f005:**
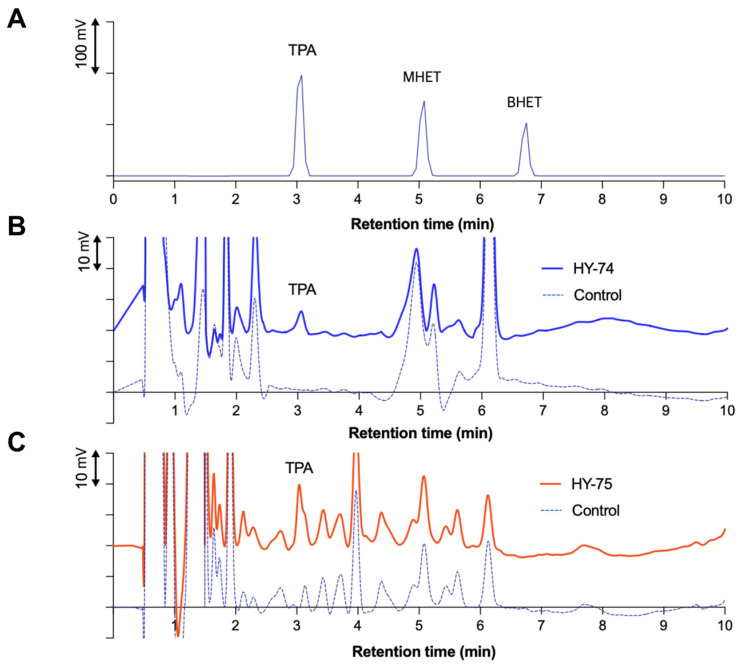
Determination of PET powder degradation by the HY-74 and HY-75 strain using high-performance liquid chromatography (HPLC) analysis after 4 weeks of incubation. (**A**) HPLC spectra of a standard sample displaying peaks for degradation products MHET, BHET, and TPA. HPLC spectra showing PET biodegradation by HY-74 strain (**B**) and HY-75 strain (**C**) after 4 weeks of incubation.

## Data Availability

The data presented in this study are available on request from the corresponding author.

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
