# Peer review of "Biodegradation Potential of Polyethylene Terephthalate by the Two Insect Gut Symbionts Xanthomonas sp. HY-74 and Bacillus sp. HY-75"

_polymers, 2023, doi:10.3390/polym15173546_

Round 1
Reviewer 1 Report
1. How many organisms were screened from the insects? How did the authors screen the two chosen organisms, HY-74 and HY-75?
2. What enzymes are responsible for plastic degradation in this study? What is the activity of these enzymes?
3. References are not relevant in many places, including [5 - 10], [20] etc.,
Author Response
Dear reviewer 1,
Thank you for your effort in reviewing our paper and for the valuable comments that have helped us to improve the manuscript. We agree with your comments and, therefore, we have revised our manuscript accordingly, as indicated in our responses to your comments listed below:
1. How many organisms were screened from the insects? How did the authors screen the two chosen organisms, HY-74 and HY-75?
In a previous study, two species of megachilid bee demonstrated the use of different types of polyurethane and polyethylene plastics as substitutes for natural materials to construct and close brood cells in nests containing successfully emerging brood (Maclvor, J.S.; Moore, A.E. Bees collect polyurethane and polyethylene plastics as novel nest materials. Ecosphere 2013, 4, 1-6). Based on this background, we hypothesize that bees (and their gut microbiota) may have the ability to degrade and utilize plastics, and indeed, through previous research, we confirmed that gut symbiont (Xanthomonas sp. HY-71) derived from Hymenoptera has a high potential for plastic degradation (Kim, J.H.; Choi, S.H.; Park, M.G.; Park, D.H.; Son, K.H.; Park, H.Y. Biodegradation of polyurethane by Japanese carpenter bee gut-associated symbionts Xanthomonas sp. HY-71, and its potential application on bioconversion. Environ. Technol. Innov. 2022, 28, 102822). Therefore, we screened PET-degrading microorganisms for the two collected species and secured only two bacteria that generate translucent halos, as shown in Fig. S1, for use in our research.
2. What enzymes are responsible for plastic degradation in this study? What is the activity of these enzymes?
In general, enzymes like lipase, cutinase, and esterase are recognized for their ability to degrade PET. These enzymes hydrolyze the ester bonds, breaking PET down into its monomer forms, MHET and BHET. Subsequently, MHET and BHET can be further broken down into TPA and EG through additional ester bond cleavage. In this study, we conducted assays for the lipase and cutinase activities on agar plates, and both extracellular reactions were confirmed. However, we fully agree with your opinion. While both extracellular reactions were confirmed, a more in-depth investigation is needed to elucidate the specific enzymes responsible for PET degradation. For the application of the two strains to real plastic waste management, it is necessary to characterize enzymes involved in the HY-74 and HY-75 strain and their metabolic pathways. Therefore, we will conduct further research on its mechanism and optimization using multi-omics and synthetic microbial communities. We described the limitations of this study and the direction of future research in the manuscript as you suggested (Lines 315-319 and 335-338).
3. References are not relevant in many places, including [5 - 10], [20] etc.,
Per your recommendation, we revised the references as you suggested (Lines 359-365, 369-371, 392-393, and 424-425).
I would like to emphasize how grateful I am for your constructive and useful comments. I hope that I have addressed your concerns and that the revised manuscript is acceptable for publication.
Yours sincerely,
Ho-Yong Park, Ph.D.

Reviewer 2 Report
Dear authors,
I have reviewed the manuscript entitled "Potential of two isolated insect gut symbionts for biodegradation of polyethylene terephthalate" and find it to be an interesting topic that would appeal to readers of the Polymer journal. I recommend minor revisions, and once the authors address these revisions thoroughly, the manuscript will be acceptable for publication. Please find my specific comments below:
1. The material and methods section in the abstract is too lengthy. I suggest summarizing it and providing more emphasis on the results.
2. It would be beneficial to include the names of the bacteria in the title for clarity.
3. It would be helpful to explain why these particular insects were chosen for the study.
4. The results should be compared with relevant studies in the field.
5. The first section of the discussion seems repetitive and could be condensed or combined with other sections.
6. It is important to mention any limitations of the study.
7. Is it possible to provide insights into the PET degradation mechanism based on the findings?
8. The conclusion should be based on the results of the study.
Author Response
Dear reviewer,
Thank you for your effort in reviewing our paper and for the valuable comments that have helped us to improve the manuscript. We agree with your comments and, therefore, we have revised our manuscript accordingly, as indicated in our responses to your comments listed below:
1. The material and methods section in the abstract is too lengthy. I suggest summarizing it and providing more emphasis on the results.
Per your recommendation, we revised the abstract as you suggested (Line 10-22).
2. It would be beneficial to include the names of the bacteria in the title for clarity.
Per your recommendation, we revised the title as you suggested (Line 2-4). “Biodegradation potential of polyethylene terephthalate by the two insect gut symbionts Xanthomonas sp. HY-74 and Bacillus sp. HY-75”
3. It would be helpful to explain why these particular insects were chosen for the study.
In a previous study, two species of megachilid bee demonstrated the use of different types of polyurethane and polyethylene plastics as substitutes for natural materials to construct and close brood cells in nests containing successfully emerging brood (Maclvor, J.S.; Moore, A.E. Bees collect polyurethane and polyethylene plastics as novel nest materials. Ecosphere 2013, 4, 1-6). Based on this background, we hypothesize that bees (and their gut microbiota) may have the ability to degrade and utilize plastics, and indeed, through previous research, we confirmed that gut symbiont (Xanthomonas sp. HY-71) derived from Hymenoptera has a high potential for plastic degradation (Kim, J.H.; Choi, S.H.; Park, M.G.; Park, D.H.; Son, K.H.; Park, H.Y. Biodegradation of polyurethane by Japanese carpenter bee gut-associated symbionts Xanthomonas sp. HY-71, and its potential application on bioconversion. Environ. Technol. Innov. 2022, 28, 102822). Therefore, we screened PET-degrading microorganisms for the two collected species and secured only two bacteria that generate translucent halos, as shown in Fig. S1, for use in our research. However, we will conduct further research (screening of various insect species including other Hymenoptera) on some limitations (only two Hymenoptera species) of this study.
4. The results should be compared with relevant studies in the field.
Per your recommendation, we revised the discussion you suggested (Line 300-303).
5. The first section of the discussion seems repetitive and could be condensed or combined with other sections.
Per your recommendation, we revised the discussion as you suggested (Line 263).
6. It is important to mention any limitations of the study.
We described the limitations of this study and the direction of future research in the manuscript in Lines 303-306, 315-319, 321-323, and 335-338.
7. Is it possible to provide insights into the PET degradation mechanism based on the findings?
We confirmed the presence of a TPA peak in the HPLC analysis, suggesting that PET may have been initially degraded to MHET or BHET and further to TPA. However, we fully agree with your opinion. The precise mechanisms underlying the degradation of TPA and EG require further investigation, as well as a deeper understanding of their metabolism including characterization of enzymes involved and their metabolic pathways. According to your opinion, we will conduct further research on some limitations of this study. We described the limitations of this study and the direction of future research in the manuscript (Line 315-319 and 335-338).
8. The conclusion should be based on the results of the study.
Per your recommendation, we revised the conclusion as you suggested (Line 331-335).
I would like to emphasize how grateful I am for your constructive and useful comments. I hope that I have addressed your concerns and that the revised manuscript is acceptable for publication.
Yours sincerely,
Ho-Yong Park, Ph.D.

Reviewer 3 Report
White pollution is serious and biodegradation of plastics is attracted. In this manuscript, polyethylene terephthalate degrading insect gut microorganisms were found and the degradation abilities etc. were investigated. The results are interesting and the writing is acceptable. It can be considered accepted after revision.
The specific comments are as follows:
(1) Hydrolysis reaction and hydrolase were inferred to be related to PET degradation. From the structure of PET and the analytic results of FTIR, if some other kinds of reactions and enzymes related to PET degradation can be inferred?
(2) PET was degraded by the living microorganisms or by the secreted enzymes (supernatant obtained by centrifugation followed by filtration by using 0.2 m pore size filter) can be further investigated. If it can be degraded by the secreted enzymes, proteome analysis of the supernatant can be made to find the possible enzymes directedly related to PET degradation. It is for your reference only, not required for this manuscript.
(3) Line 285, “grading” changes to “degrading”.
The English is acceptable.
Author Response
Dear reviewer,
Thank you for your effort in reviewing our paper and for the valuable comments that have helped us to improve the manuscript. We agree with your comments and, therefore, we have revised our manuscript accordingly, as indicated in our responses to your comments listed below:
1. Hydrolysis reaction and hydrolase were inferred to be related to PET degradation. From the structure of PET and the analytic results of FTIR, if some other kinds of reactions and enzymes related to PET degradation can be inferred?
C-O stretching (1,097 cm-1) indicates ester linkages between terephthalate and EG units. Furthermore, the C-O-C stretching (1,240 cm-1) represents the vibrations in the ester group, and C=O stretching (1,714 cm-1) signifies the cleavage of the ester bonds in PET. The increased hydroxyl group (OH) peak (3,200–3,600 cm-1) suggested the formation of OH in the degradation products. These peaks might indicate ester bond cleavage and formation of EG. The precise mechanisms underlying the degradation of TPA and EG require further investigation, as well as a deeper understanding of their metabolism including characterization of enzymes involved and their metabolic pathways. According to your opinion, we will conduct further research on some limitations of this study. We described the limitations of this study and the direction of future research in the manuscript (Line 315-319 and 335-338).
2. PET was degraded by the living microorganisms or by the secreted enzymes (supernatant obtained by centrifugation followed by filtration by using 0.2 m pore size filter) can be further investigated. If it can be degraded by the secreted enzymes, proteome analysis of the supernatant can be made to find the possible enzymes directedly related to PET degradation. It is for your reference only, not required for this manuscript.
Due to the prolonged time required for PET degradation, it's also challenging to perform proteome analysis on the supernatant. However, we agree with your opinion. While both extracellular reactions were confirmed in agar plate condition, it is necessary to characterize enzymes involved in the HY-74 and HY-75 strain and their metabolic pathways. A more in-depth investigation is needed to elucidate the specific enzymes responsible for PET degradation. Further, we will conduct further research on its mechanism and optimization using multi-omics and synthetic microbial communities. According to your opinion, we will conduct further research on some limitations of this study. We described the limitations of this study and the direction of future research in the manuscript as you suggested (Lines 315-319 and 335-338).
3. Line 285, “grading” changes to “degrading”.
Per your recommendation, we revised the manuscript as you suggested (Line 280).
I would like to emphasize how grateful I am for your constructive and useful comments. I hope that I have addressed your concerns and that the revised manuscript is acceptable for publication.
Yours sincerely,
Ho-Yong Park, Ph.D.

Reviewer 4 Report
Two isolated strains were used for PET biodegradation, which is hotpoint recently. However, it is difficult for me (or readers) to understand what idea authors want to express in the manuscript, because it is similar to a experimental report because concerned mechanisms were not explained clearly, and experimental methods were not described clearly. For example, "the flasks were incubated on a rotary shaker (Line 125)", for which can the flask be incubated? How to prepare fresh media?
In addition, authors did not explain what meaning of Fig. 3G and Fig. 3F, for which I did not the what authors want to experss. What is difference between PET film and powder? According to authors' description, powder could be disolved in medium, for which I wonder it is true? If it is not true, how did authors measure PET in HPLC?
So many similar questions could not be explained clearly, which let me not recommend it for publication in Polymers.
Writing in English was poor.
Author Response
Dear reviewer,
Thank you for your effort in reviewing our paper and for the valuable comments that have helped us to improve the manuscript. We agree with your comments and, therefore, we have revised our manuscript accordingly, as indicated in our responses to your comments listed below:
1. Two isolated strains were used for PET biodegradation, which is hotpoint recently. However, it is difficult for me (or readers) to understand what idea authors want to express in the manuscript, because it is similar to a experimental report because concerned mechanisms were not explained clearly.
We confirmed the presence of a TPA peak in the HPLC analysis, suggesting that PET may have been initially degraded to MHET or BHET and further to TPA. However, we fully agree with your opinion. The precise mechanisms underlying the degradation of TPA and EG require further investigation, as well as a deeper understanding of their metabolism including characterization of enzymes involved and their metabolic pathways. According to your opinion, we will conduct further research on some limitations of this study. We described the limitations of this study and the direction of future research in the manuscript (Line 315-319 and 335-338).
Experimental methods were not described clearly. For example, "the flasks were incubated on a rotary shaker (Line 125)", for which can the flask be incubated? How to prepare fresh media?
Per your recommendation, we revised the method as you suggested (Line 122-126).
2. In addition, authors did not explain what meaning of Fig. 3G and Fig. 3F, for which I did not the what authors want to experss. What is difference between PET film and powder? According to authors' description, powder could be disolved in medium, for which I wonder it is true? If it is not true, how did authors measure PET in HPLC?
PET film powder cannot be dissolved in water due to its hydrophobic chemical properties, but it can be distributed more evenly in bacterial culture compared to PET film, offering a larger surface area. We examined the degradation products of PET, namely MHET and TPA, which are slightly soluble in water. Per your recommendation, we revised the figure descriptions for Figures 3E and 3F (Line 207-209).
I would like to emphasize how grateful I am for your constructive and useful comments. I hope that I have addressed your concerns and that the revised manuscript is acceptable for publication.
Yours sincerely,
Ho-Yong Park, Ph.D.

Round 2
Reviewer 4 Report
Authors revised their manuscript according to comments from reviewers, and it reaches at the level, and I recommend it publication.